# Comparison of Lower Limb COP and Muscle Activation During Single-Leg Deadlift Using Elastic and Inelastic Barbells

**DOI:** 10.3390/sports13080242

**Published:** 2025-07-24

**Authors:** Jihwan Jeong, Ilbong Park

**Affiliations:** Department of Sports Health Convergence, Busan University of Foreign Studies, Busan 46234, Republic of Korea; fnjboss@bufs.ac.kr

**Keywords:** single-leg deadlift, COP, EMG, elastic barbell, inelastic barbell, sports rehabilitation, neuromuscular control

## Abstract

Background: This study aimed to investigate how barbell type (elastic vs. inelastic) and lifting speed affect postural stability and lower limb muscle activation during the single-leg deadlift (SLDL), a common unilateral exercise in rehabilitation and performance training. Methods: Twenty-seven healthy adults performed SLDLs using both elastic and inelastic barbells under three lifting speeds (normal, fast, and power). Center of pressure (COP) displacement in the anterior–posterior (AP) and medial–lateral (ML) directions and electromyographic (EMG) activity of eight lower limb muscles were measured. Results: COP displacement was significantly lower when using elastic barbells (AP: F = 6.509, *p* = 0.017, η^2^ = 0.200, ω^2^ = 0.164; ML: F = 9.996, *p* = 0.004, η^2^ = 0.278, ω^2^ = 0.243). EMG activation was significantly higher for the gluteus medius, biceps femoris, semitendinosus, and gastrocnemius (all *p* < 0.01), especially under power conditions. Significant interactions between barbell type and speed were found for the gluteus medius (F = 13.737, *p* < 0.001, η^2^ = 0.346, ω^2^ = 0.176), semitendinosus (F = 6.757, *p* = 0.002, η^2^ = 0.206, ω^2^ = 0.080), and tibialis anterior (F = 3.617, *p* = 0.034, η^2^ = 0.122, ω^2^ = 0.029). Conclusions: The findings suggest that elastic barbells improve postural control and enhance neuromuscular activation during the SLDL, particularly at higher speeds. These results support the integration of elastic resistance in dynamic balance and injury prevention programs.

## 1. Introduction

The Single-Leg Deadlift (SLDL) is widely utilized for enhancing lower limb strength and stability [1]. As a form of closed kinetic chain exercise, it primarily targets the hamstrings and has been proven effective in reducing the risk of hamstring injuries [2,3]. In sports rehabilitation, the SLDL is recommended for strengthening the lumbar–pelvic musculature and posterior chain muscles [4]. It is considered a functional compound movement rather than an isolated joint exercise, emphasizing neuromuscular coordination over selective muscle strengthening [5,6].

The gluteus medius plays a crucial role in maintaining knee alignment [7]. Impaired activation of this muscle may lead to increased genu valgum, knee pain, instability, and heightened risk of anterior cruciate ligament (ACL) injury [8]. Among ACL injury prevention exercises, the SLDL is particularly effective, promoting balanced strengthening of both hamstrings and quadriceps femoris muscles [9].

The SLDL enhances neuromuscular activation in the lower extremities [10], particularly hamstrings, which contribute to dynamic knee stability through co-contraction with quadriceps and reducing anterior shear forces on the ACL [11,12]. Eccentric contractions of the hamstrings during the SLDL can reportedly reduce injury incidence by up to 51% and are considered one of the fastest ways to improve hamstring strength [13,14].

Recent studies have advocated increasing the use of unilateral training, such as the SLDL, grounded in the principle of sport-specific functional training [15,16]. Such training enhances knee function, muscle strength, joint stability, coordination, and neuromuscular control [17,18]. Previous comparisons between unilateral and bilateral leg training have shown equal or superior outcomes in unilateral protocols [19,20,21]. Furthermore, unilateral exercises reduce spinal compression, increase joint stability across multiple planes, and improve proprioceptive responses to perturbation [22,23,24,25].

Although traditionally not recommended for pure strength development, instability tools are gaining popularity as methods to enhance neuromuscular adaptations in resistance training [26,27]. These tools are particularly effective in performance enhancement and injury prevention among athletes by increasing activation of stabilizing muscles in unstable environments [28,29]. While bottom-up instability methods have been extensively studied, fewer investigations have focused on top-down instability, where instability originates in the upper limbs and transfers to the lower limbs [30,31].

Elastic barbells, made of flexible plastic materials, are now being introduced as resistance tools in practice. Their inherent elasticity creates dynamic perturbations during movements, which may enhance joint stability and coordination [32,33]. Recent findings have shown that American football players exhibited significant differences in lower limb electromyographic (EMG) and ground reaction forces (GRFs) during squats with elastic versus steel barbells [34].

Based on the principles of sport-specific training, using alternative forms of environmental instability, such as elastic bars, may be more effective than traditional unstable surfaces [35]. Since EMG provides insight into muscular force production during sports movement [36], it is a valuable tool for assessing the physiological impact of such training modalities [37].

Despite increasing field applications, research on elastic barbells especially involving top-down instability is still limited. In the context of modern rehabilitation and athletic performance enhancement, the SLDL has garnered increasing attention due to its capacity to improve unilateral balance, posterior chain engagement, and proprioceptive function. However, while the biomechanical and neuromuscular benefits of the SLDL have been documented in stable environments [3], less is known about how equipment-induced instability particularly top-down perturbation via elastic barbells affects postural control and muscle coordination. Unlike traditional unstable surfaces, elastic barbells introduce dynamic fluctuations during movement that challenge kinetic chain integrity from proximal to distal joints [27].

Recent studies suggest that such instability can alter central nervous system recruitment strategies, enhancing motor unit synchronization and joint stabilization [26]. Moreover, these perturbations may increase eccentric loading demands, thereby improving muscular strength and injury resilience, especially in muscles prone to strain injuries such as the hamstrings and the gluteus medius [13]. Despite these theoretical advantages, there remains a paucity of evidence evaluating elastic barbells in functional exercises like the SLDL, especially regarding how their use influences balance and lower limb neuromuscular activation across different movement speeds.

Therefore, this study aimed to investigate how different barbell types (elastic vs. inelastic) and lifting speeds (normal, fast, and power) influence center of pressure (COP) displacement and lower limb muscle activation during SLDL performance. By addressing the underexplored impact of top-down instability through elastic barbell use, this study seeks to provide biomechanical insights into postural control and neuromuscular coordination strategies during dynamic unilateral tasks.

We hypothesized that elastic barbells would result in reduced COP displacement and increased muscle activation compared to inelastic barbells, particularly under high-velocity conditions.

## 2. Materials and Methods

### 2.1. Participants

A total of 27 healthy adults (Age: 21.9 ± 1.6 years; height: 177.6 ± 4.5 cm; weight: 76.6 ± 8.9 kg) with at least one year of resistance training experience and no musculoskeletal disorders were recruited from the Lifelong Education Center of B University. The study protocol was approved by the Institutional Review Board (Approval No: P01-202502-01-027). All participants provided informed consent after being briefed about the study’s aims and procedures. The target sample size for this study was determined using G*Power 3.1. Based on prior studies that investigated biomechanical and neuromuscular characteristics during single-leg deadlift performance [1,38,39], the effect size was set to 0.3. With an alpha level of 0.05 and statistical power of 0.80, the minimum required number of participants was calculated to be 24. However, considering the technical difficulty of the single-leg deadlift and the anticipated dropout rate of approximately 15–20%, a total of 30 participants were recruited to ensure adequate statistical power.

### 2.2. Study Design

This study was designed as a randomized crossover trial, in accordance with the CONSORT extension for crossover designs. Each participant performed the single-leg deadlift (SLDL) using both elastic and inelastic barbells under three lifting speeds (normal, fast, and power). To mitigate potential order effects, the initial barbell condition was randomly allocated using a computer-generated sequence, and a one-week washout period was implemented between sessions.

Due to the visible characteristics of the barbells, blinding of participants and assessors was not feasible. However, data coding and statistical analysis were conducted by an independent investigator blinded to group allocation. Although allocation concealment was not applicable in a traditional sense due to the crossover design, procedural consistency was ensured across sessions.

This within-subject crossover design increases the statistical power by reducing inter-individual variability. All participants were familiarized with the exercise protocols before testing to minimize learning effects. Ethical approval was obtained, and informed consent was secured from all participants in accordance with institutional guidelines.

### 2.3. Measurement Tools and Procedures

#### 2.3.1. COP and EMG Assessment

Participants performed the SLDL using both elastic and inelastic barbells, each set at 30% of their previously assessed one-repetition maximum (1RM) [34]. The 1RM was estimated one week prior using the 7–10 repetition submaximal test to ensure safety [40]. All movements were conducted using the dominant leg. COP was measured using a force platform (AMTI-OR6, AMTI Watertown, MA, USA) at a sampling frequency of 1000 Hz. A minimum threshold of 10 N was set to define valid data, and non-slip tape was applied to the force plate to prevent slippage. To eliminate electrical noise, equipment was pre-warmed for at least 45 min before measurements [41].

Surface EMG was used to record muscle activity using wireless EMG equipment (Ultium system, Noraxon Inc., Scottsdale, AZ, USA). Bipolar Ag/AgCl surface electrodes were placed on the following muscles: gluteus medius, rectus femoris, vastus medialis, vastus lateralis, biceps femoris, semitendinosus, gastrocnemius, and tibialis anterior, following SENIAM guidelines (Figure 1) [42].

EMG data were normalized using the percentage of Maximum Voluntary Isometric Contraction (%MVIC). Each MVIC was measured over three 5 s contractions, excluding the first and last seconds. Standardization followed the Manual Muscle Testing (MMT) protocol [43]. Electrodes were placed on the muscle belly with the active and reference electrodes positioned parallel to muscle fibers, spaced 1 cm apart [44]. Measurement equipment is listed in Table 1.

#### 2.3.2. Data Collection Protocol

Participants wore shorts and T-shirts and performed exercises barefoot. They received more than 20 min of training before measurement to ensure accuracy and safety. Each SLDL trial included a concentric lifting phase, a 2 s standing balance phase, and an eccentric lowering phase. The lifting phase was executed at three speeds—normal (2 s), fast (1 s), and power (as fast as possible). The barbell order was randomized, and each condition was performed three times. A metronome was used to control timing, and participants were familiarized with its rhythm in advance. A 5 min rest was given between sets to prevent fatigue. Trials were repeated if balance was lost or the foot was displaced from the force platform. EMG and COP data were synchronized using a Vicon motion capture system (MX-T20, Vicon, Oxford, UK) (Figure 2 and Figure 3).

#### 2.3.3. Data Processing and Analysis

COP data were processed using Nexus software (Nexus v2.15, Vicon, Oxford, UK) and analyzed for anterior–posterior (AP) and medial–lateral (ML) displacement ranges. EMG data were analyzed using MR 3.20 (Noraxon, Scottsdale, AZ, USA) with a 2000 Hz sampling frequency and a band-pass filter of 20–400 Hz. The root mean square (RMS) window was set at 150 ms. Data were normalized to MVIC.

### 2.4. Statistical Analysis

All data were analyzed using SPSS 22.0 (IBM Corp., Armonk, NY, USA). Means and standard deviations were calculated. A two-way repeated measures ANOVA was performed to examine the effects of barbell type and lifting speed on COP and EMG variables. Bonferroni correction was used for post hoc analyses, and statistical significance was set at α = 0.05.

## 3. Results

### 3.1. Center of Pressure (COP)

The two-way repeated measures ANOVA revealed significant main effects of barbell type and speed on COP. In the anterior–posterior (AP) direction, elastic barbells showed significantly lower displacement than inelastic barbells (F = 6.509, *p* = 0.017, η^2^ = 0.200, ω^2^ = 0.164), and significant differences were observed across speed conditions (F = 3.984, *p* = 0.025, η^2^ = 0.133, ω^2^ = 0.052). Similarly, in the medial–lateral (ML) direction, elastic barbells resulted in reduced displacement compared to inelastic barbells (F = 9.996, *p* = 0.004, η^2^ = 0.278, ω^2^ = 0.243), with speed also producing a significant main effect (F = 18.330, *p* < 0.001, η^2^ = 0.413, ω^2^ = 0.243). However, no significant interaction effects between barbell type and speed were found for either AP or ML directions. As shown in Table 2, COP displacement in both AP and ML directions was significantly lower when using elastic barbells, particularly under power-speed conditions.

### 3.2. Electromyography (EMG)

EMG analysis indicated that muscle activation in the gluteus medius (GMed), biceps femoris (BF), semitendinosus (ST), and gastrocnemius (GCN) was significantly higher when using elastic barbells compared to inelastic barbells (GMed: F = 38.02, *p* < 0.001, η^2^ = 0.594, ω^2^ = 0.569; BF: F = 21.976, *p* < 0.001, η^2^ = 0.458, ω^2^ = 0.414; ST: F = 15.165, *p* = 0.001, η^2^ = 0.369, ω^2^ = 0.320; GCN: F = 13.063, *p* = 0.001, η^2^ = 0.334, ω^2^ = 0.281). Additionally, lifting speed had a significant main effect across all measured muscles (*p* < 0.001). Notably, interaction effects between barbell type and speed were significant for GMed (F = 13.737, *p* < 0.001, η^2^ = 0.346, ω^2^ = 0.176), ST (F = 6.757, *p* = 0.002, η^2^ = 0.206, ω^2^ = 0.080), and TA (F = 3.617, *p* = 0.034, η^2^ = 0.122, ω^2^ = 0.029), with the greatest activation observed in the power condition using the elastic barbell. According to Table 3, EMG activity of the gluteus medius and semitendinosus increased significantly with elastic barbell use, and the interaction between barbell type and speed was statistically significant.

## 4. Discussion

This study aimed to investigate the effects of barbell type (elastic vs. inelastic) and lifting speed on COP displacement and lower extremity muscle activation during SLDL exercises. The findings suggest that the use of elastic barbells significantly influences both postural control and neuromuscular activation patterns, with important implications for athletic training and rehabilitation programs.

The results demonstrated that using an elastic barbell significantly reduced COP displacement in both AP and ML directions compared to the inelastic barbell. These findings align with previous research indicating that instability-based resistance training enhances neuromuscular control and dynamic balance [45,46]. The reduction in COP displacement may reflect improved postural stability due to enhanced activation of stabilizing muscles in response to top-down perturbations introduced by the elastic barbell.

Particularly in the power condition, ML displacement increased by up to 18.3%, suggesting that high-velocity movements amplify the body’s adaptive strategies to maintain balance. This supports the findings of Mansfield et al. [32], who reported that increased COP variability under unstable conditions enhances proprioceptive feedback. These results also validate the concept of top-down instability, where perturbations initiated at the upper limbs propagate through the kinetic chain to influence lower limb mechanics.

EMG analysis revealed that activation of the gluteus medius, biceps femoris, semitendinosus, and gastrocnemius was significantly greater when using elastic barbells, particularly at maximal movement speed. Increased gluteus medius activation (e.g., 48.3% MVIC in power condition) indicates enhanced hip joint stabilization, which may reduce the risk of knee valgus, a known contributor to ACL injuries [7]. This supports the use of elastic barbells as a means to improve hip muscle engagement and lower limb alignment during functional tasks.

The co-activation of biceps femoris and semitendinosus observed in the elastic barbell condition reflects increased hamstring recruitment, particularly under high-speed loading. This neuromuscular pattern plays a key role in ACL protection by countering anterior tibial translation [12,13]. The findings suggest that elastic barbell resistance training may be a practical approach to reducing ACL injury risk, especially in sports requiring explosive single-leg movements.

Furthermore, the observed increase in tibialis anterior and gastrocnemius activation under elastic barbell and power-speed conditions indicates heightened demand for ankle stabilization, supporting the concept of an ankle strategy in balance control. This suggests that perturbations originating from the upper body effectively challenge and train distal joint stability through the kinetic chain [25].

The enhanced muscle activation in the power condition across all measured muscles also reflects a greater neuromuscular demand, consistent with previous findings that velocity influences γ-motor neuron sensitivity and increases muscle spindle responsiveness [10]. This physiological response is critical for dynamic joint stability during high-speed movements.

The results further suggest that elastic barbells may serve as an effective training tool for improving joint stabilization, proprioceptive response, and coordination in athletic and rehabilitative settings. As reported by Park et al. [47], interventions incorporating elastic barbell-induced instability can enhance neuromuscular control and reduce injury incidence.

In designing this crossover trial, careful attention was given to maintaining equity in training load (TL) across all experimental conditions. This approach ensured that observed differences in postural control and muscle activation were attributable to the intervention factors barbell type and lifting speed rather than imbalanced neuromuscular demands. Recent recommendations in sports science emphasize the importance of standardizing TL when assessing intervention efficacy, especially in crossover or repeated-measures designs [48]. This methodological consideration enhances the internal validity of the current findings and strengthens their translational relevance in athletic and rehabilitation settings.

However, this study has several limitations. First, the sample was limited to healthy young adult males, limiting the generalizability of findings to other populations. Future studies should include diverse age groups and female participants. Second, individual variations in movement technique may have influenced EMG and COP results, despite the standardization and familiarization procedures. Third, this was a short-term cross-sectional study; thus, longitudinal studies are needed to investigate the long-term effects of elastic barbell training on strength and joint stability.

Additionally, future studies may consider incorporating near-infrared spectroscopy (NIRS) to assess local muscle oxygenation and metabolic responses. This could complement EMG findings and provide a deeper insight into the physiological demands of elastic versus inelastic resistance exercises [49].

## 5. Conclusions

This study demonstrated that using elastic barbells during SLDL leads to significantly reduced COP displacement and increased lower limb muscle activation compared to inelastic barbells. These effects were particularly pronounced at higher movement speeds, supporting our initial hypothesis.

The findings suggest that elastic barbells are effective in enhancing dynamic postural control and neuromuscular coordination, making them a practical tool for both athletic performance development and injury prevention in rehabilitation settings. Given the growing interest in top-down instability modalities, elastic barbell training may serve as a functional alternative to traditional instability devices.

Future studies should examine long-term adaptations to elastic resistance training, assess biomechanical and metabolic responses using tools such as NIRS and include broader participant demographics to increase generalizability.

## Figures and Tables

**Figure 1 sports-13-00242-f001:**
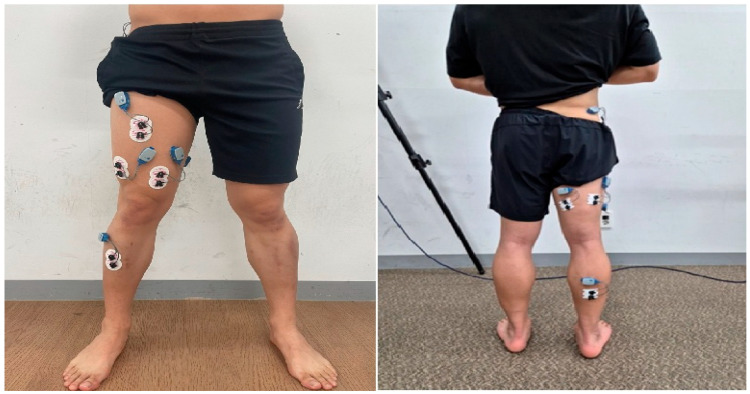
Surface electromyography attachment sites.

**Figure 2 sports-13-00242-f002:**
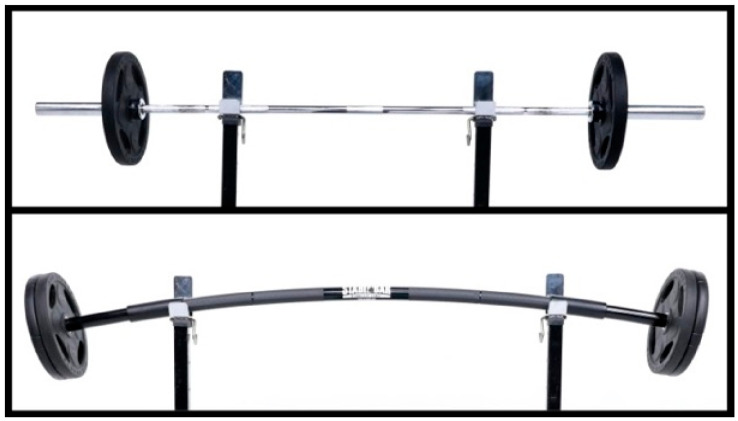
The upper image shows the inelastic barbell, while the lower image shows the elastic barbell.

**Figure 3 sports-13-00242-f003:**
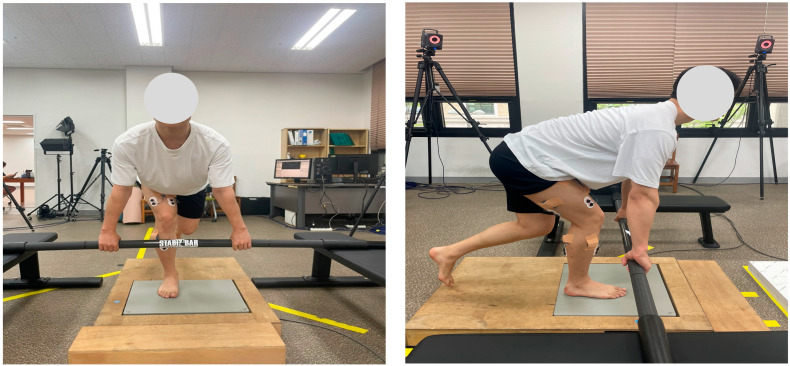
The subject performed a single-leg deadlift movement under both conditions.

**Table 1 sports-13-00242-t001:** COP and EMG measurement device.

Equipment	Model	Manufacture
Force platform	OR6	AMTI, MA, USA
COP analysis software	Nexus v2.15	Vicon, Oxford, UK
EMG equipment	Ultium System	Noraxon, AZ, USA
EMG analysis software	MR 3.20	Noraxon, AZ, USA
3D motion capture	MX-T20	Vicon, Oxford, UK

**Table 2 sports-13-00242-t002:** Comparison of center of pressure (COP) displacement in anterior–posterior (AP) and medial–lateral (ML) directions according to barbell type and movement speed during single-leg deadlift (mean ± SD, partial η^2^, ω^2^).

		Inelastic	Elastic	Source	F	*p*	η^2^	ω^2^	Bonferroni
AP	Normal ^a^	0.95 ± 0.21	0.90 ± 0.17	Type	6.509	0.017 *	0.200	0.164	a < b *
	Fast ^b^	0.85 ± 0.26	0.81 ± 0.18	Speed	3.984	0.025 *	0.133	0.052	
	Power ^c^	0.91 ± 0.16	0.82 ± 0.13	Type × Speed	0.978	0.383	0.036	0.000	
ML	Normal ^a^	1.23 ± 0.20	1.18 ± 0.20	Type	9.996	0.004 **	0.278	0.243	a < c *** b < c **
	Fast ^b^	1.21 ± 0.22	1.12 ± 0.25	Speed	18.330	0.000 ***	0.413	0.243	
	Power ^c^	1.10 ± 0.12	0.99 ± 0.10	Type × Speed	0.925	0.403	0.034	0.000	

Note: Data are presented as mean ± standard deviation. AP: anterior–posterior; ML: medial–lateral. a: normal speed; b: fast speed; c: power speed (* *p* < 0.05, ** *p* < 0.01, *** *p* < 0.001).

**Table 3 sports-13-00242-t003:** Comparison of root mean square (RMS) electromyographic activity (%MVIC) of lower limb muscles under different barbell types and movement speeds during single-leg deadlift (mean ± SD, partial η^2^, ω^2^).

		Inelastic	Elastic	Source	F	*p*	η^2^	ω^2^	Bonferroni
GMed	Normal ^a^	30.93 ± 10.82	32.16 ± 11.74	Type	38.02	0.000 ***	0.594	0.569	a < b < c
	Fast ^b^	32.94 ± 11.96	35.54 ± 12.51	Speed	64.327	0.000 ***	0.712	0.531	
	Power ^c^	42.73 ± 16.69	48.33 ± 17.57	Type × Speed	13.737	0.000 ***	0.346	0.176	
RF	Normal ^a^	5.98 ± 3.21	5.79 ± 2.80	Type	2.035	0.166	0.073	0.036	a < c b < c
	Fast ^b^	6.38 ± 3.01	6.61 ± 3.24	Speed	101.176	0.000 ***	0.796	0.643	
	Power ^c^	12.12 ± 4.69	12.98 ± 5.73	Type × Speed	1.352	0.260	0.049	0.000	
VM	Normal ^a^	14.23 ± 5.89	13.63 ± 6.16	Type	0.028	0.869	0.001	0.000	a < c b < c
	Fast ^b^	14.95 ± 5.87	14.47 ± 5.61	Speed	74.909	0.000 ***	0.742	0.570	
	Power ^c^	24.25 ± 10.27	25.13 ± 9.71	Type × Speed	1.534	0.229	0.056	0.000	
VL	Normal ^a^	19.05 ± 7.06	18.03 ± 6.94	Type	0.497	0.487	0.019	0.000	a < c b < c
	Fast ^b^	19.48 ± 6.82	18.86 ± 6.72	Speed	104.290	0.000 ***	0.800	0.650	
	Power ^c^	28.96 ± 8.99	29.75 ± 9.40	Type × Speed	2.561	0.106	0.090	0.010	
BF	Normal ^a^	23.69 ± 9.76	24.58 ± 8.76	Type	21.976	0.000 ***	0.485	0.428	a < c b < c
	Fast ^b^	23.58 ± 10.00	26.29 ± 10.65	Speed	30.430	0.000 ***	0.539	0.341	
	Power ^c^	27.46 ± 9.73	30.14 ± 12.05	Type × Speed	1.721	0.194	0.062	0.000	
ST	Normal ^a^	24.21 ± 9.92	24.47 ± 9.42	Type	15.165	0.001 **	0.368	0.336	a < b < c
	Fast ^b^	24.83 ± 10.19	26.86 ± 10.50	Speed	56.494	0.000 ***	0.685	0.498	
	Power ^c^	31.58 ± 13.08	34.69 ± 14.79	Type × Speed	6.757	0.002 **	0.206	0.080	
GCN	Normal ^a^	32.82 ± 11.17	33.34 ± 9.48	Type	13.063	0.001 **	0.334	0.301	a < b < c
	Fast ^b^	34.23 ± 11.29	36.73 ± 13.95	Speed	91.073	0.000 ***	0.778	0.618	
	Power ^c^	45.49 ± 15.08	49.62 ± 16.11	Type × Speed	3.135	0.052	0.108	0.020	
TA	Normal ^a^	24.56 ± 9.13	22.77 ± 7.41	Type	0.289	0.595	0.011	0.000	a < c
	Fast ^b^	25.08 ± 8.43	25.65 ± 8.59	Speed	146.662	0.000 ***	0.849	0.725	
	Power ^c^	38.28 ± 9.16	40.72 ± 11.64	Type × Speed	3.617	0.034 *	0.122	0.029	

Note. RMS EMG values are expressed as mean ± standard deviation (SD), normalized to %MVIC. Abbreviations: gluteus medius (GMed), rectus femoris (RF), vastus medialis (VM), vastus lateralis (VL), biceps femoris (BF), semitendinosus (ST), gastrocnemius (GCN), tibialis anterior (TA). a: normal speed; b: fast speed; c: power speed (* *p* < 0.05, ** *p* < 0.01, *** *p* < 0.001).

## Data Availability

The data used in this study are available upon reasonable request and will be deposited in a public repository upon publication.

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
