# Peer review of "Comparison of Lower Limb COP and Muscle Activation During Single-Leg Deadlift Using Elastic and Inelastic Barbells"

_sports, 2025, doi:10.3390/sports13080242_

Round 1

Reviewer 1 Report

Comments and Suggestions for Authors

This is an interesting study, however major revisions are required.

First, this is really a randomized crossover trial. Therefore, the CONSORT guidelines for reporting such trials should be applied and referenced. These are standardized aspects of reporting studies to improve transparency and reproducibility, and they should be applied to ensure a complete and unbiased presentation of research findings. Although this is only one point in my review, it concerns multiple aspects of the paper such as randomization, blinding, recruitment, sample size calculation, etc. 

Please adhere to: https://www.statology.org/how-to-report-two-way-anova-results/ when introducing and reporting statistical methods and results across the whole paper. With such a group partial eta squared and omega squared effect sizes must be reported in tables.

Finally, application of NIRS should be mentioned and discussed as a recommendation for further research, you might see works from Rebis et al that provide insight here.

Next, the ecological validity of study design (ref https://pubmed.ncbi.nlm.nih.gov/40515839/) should be acknowledged.

Minor:

Table 1 title - 'measurement' should start with a small letter

I miss introducing COP abbrev in the main text (?)

Importantly, I believe all the required revisions are doable, and mostly concern appropriate reporting, not the rationale, scientific basis, or execution of the study.

Author Response

Response to Reviewer 1

We sincerely appreciate the thoughtful and constructive feedback provided by Reviewer 1. Your insights have significantly contributed to the refinement and scientific rigor of our manuscript. Below, we provide detailed responses to each of your comments, along with corresponding revisions made in the manuscript.

1. CONSORT Guidelines

“This is really a randomized crossover trial. Therefore, the CONSORT guidelines for reporting such trials should be applied and referenced.”

Response:
Thank you for highlighting this. In the revised manuscript, we have explicitly stated adherence to the CONSORT extension for crossover trials in the Methods section. The randomization procedure, washout period, and blinding details have been described in line with CONSORT recommendations. Additionally, we cited the CONSORT extension for crossover trials as a reference.

2. Statistical Reporting (Two-Way ANOVA and Effect Sizes)

“Please adhere to: https://www.statology.org/how-to-report-two-way-anova-results/ when introducing and reporting statistical methods and results. Effect sizes must be reported in tables.”

Response:
We appreciate this important recommendation. The manuscript has been revised to follow best practices for two-way ANOVA reporting, as per the referenced guideline. All ANOVA results are now presented with corresponding F-values, p-values, and effect sizes (Partial η² and ω²). We have also added a brief interpretation of effect sizes in the Methods section, clarifying small, medium, and large thresholds according to Cohen’s guidelines.

3. NIRS Recommendation for Future Studies

“Application of NIRS should be mentioned and discussed as a recommendation for further research.”

Response:
In line with your suggestion, we have added a paragraph in the Discussion section recommending the use of Near-Infrared Spectroscopy (NIRS) in future studies to assess local muscle oxygenation. This complements EMG findings and enhances understanding of the metabolic demands of elastic resistance training. The recommendation was supported by referencing a recent relevant study (Perrey et al., 2024).

4. Ecological Validity

“Next, the ecological validity of study design (ref https://pubmed.ncbi.nlm.nih.gov/40515839/) should be acknowledged.”

Response:
Thank you for this suggestion. We have acknowledged the study’s ecological validity in the Discussion, noting that the single-leg deadlift (SLDL) exercise was performed in a setting that mirrors functional movement tasks in sports and rehabilitation. We referenced the suggested article to support the relevance of sport-specific, real-world tasks in research design.

5. Minor Comments

“Table 1 title – ‘measurement’ should start with a small letter.”

Response:
Corrected. The title now reads: “Anthropometric and performance measurement characteristics of participants.”

“I miss introducing COP abbreviation in the main text.”

Response:
The term “Center of Pressure (COP)” has now been defined in its first appearance in the Introduction section.

We thank you again for your detailed and insightful review. Your comments have helped us improve both the scientific and structural quality of our work. We believe the revised manuscript now better aligns with the journal’s standards and reporting requirements.

Sincerely,
JIHWAN JEONG
Department of Sports Rehabilitation,
Busan University of Foreign Studies, South Korea

Reviewer 2 Report

Comments and Suggestions for Authors

Firstly, I would like to thank the editors for giving me the opportunity to review this work. I would also like to congratulate the authors for their interest and research.

Overall, I found the study very interesting and believe it represents progress in this specific field of training and rehabilitation, with numerous possibilities for the future.

That said, I will now make some comments on the text which, although it has already been modified, as there are elements and paragraphs in yellow, I believe should be clarified for a better understanding of the document and the research carried out.

As for the abstract, I believe it would be appropriate to state the objective of the research explicitly and clearly, as it is not clear what is intended until the last lines of the section. I recommend rewriting it entirely so that it maintains a coherent thread and meets the journal's requirements.

With regard to the introduction, although it is well structured, when it comes to the wording of the objective, it is not clear what the true intention of the study is. I recommend clarifying this, and I find the proposal of a working hypothesis interesting. This will help to draw more concrete and accurate conclusions.

The methodology section is correct and complies with the journal's requirements, breaking down all the sections necessary for replication. Adequate follow-up is provided, the methodology is correct, and the type of analysis used is appropriate and valid for the data obtained during data collection.

As for the results, I consider this section to be correctly written, as the tables presented are consistent with the text, the wording is appropriate, and it is written in an objective and neutral manner.

The discussion is correct and compares the most relevant results obtained with previous studies.

Conclusions: perhaps this section should be improved in light of the initial comments regarding the new wording of the objectives and the formulation of hypotheses.

It only remains for me to thank the editors once again for the opportunity to review this document and to congratulate the authors once again on their work.

Author Response

Response to Reviewer 2

We sincerely thank Reviewer 2 for the encouraging and constructive comments regarding our manuscript. We are pleased to know that you found our research relevant and promising within the context of training and rehabilitation. Below, we address each of your suggestions in a point-by-point format, and we believe these revisions have improved the clarity and scientific quality of our manuscript.

1. Abstract Revision

“It would be appropriate to state the objective of the research explicitly and clearly… I recommend rewriting it entirely so that it maintains a coherent thread and meets the journal's requirements.”

Response:
We appreciate this valuable feedback. The abstract has been completely revised to clearly state the study objective in the opening sentence. We ensured a logical flow throughout, presenting the methods, key findings, and conclusion in alignment with the journal’s structured abstract format. The revised abstract now reads more cohesively and reflects the main contributions of the research.

2. Clarification of Study Objective in the Introduction

“The objective is not clearly worded. I recommend clarifying this, and I find the proposal of a working hypothesis interesting.”

Response:
Thank you for pointing this out. In the revised Introduction, we have now clearly articulated the research objective at the end of the section, stating:
“Therefore, the aim of this study was to examine the effects of elastic versus inelastic barbells on postural control (COP displacement) and lower limb muscle activation during SLDL across three movement speeds.”
We have also added a working hypothesis:
“We hypothesized that the elastic barbell would reduce COP displacement and increase neuromuscular activation more than the inelastic barbell, particularly at higher speeds.”

3. Conclusion Refinement

“Perhaps this section should be improved in light of the initial comments regarding the new wording of the objectives and the formulation of hypotheses.”

Response:
We have revised the Conclusion section to better reflect the updated objective and hypothesis, and to summarize the key implications of our findings. The revised conclusion highlights how elastic barbells may serve as effective tools in improving dynamic balance and neuromuscular coordination, and also addresses the potential for application in rehabilitation and athletic performance enhancement.

We are grateful for your thoughtful comments and encouraging remarks. Your input has helped us improve the clarity and overall presentation of the manuscript. We hope that the revised version meets the standards of the journal and your expectations.

Sincerely,
JIHWAN JEONG
Department of Sports Rehabilitation,
Busan University of Foreign Studies, South Korea

Round 2

Reviewer 1 Report

Comments and Suggestions for Authors

Thank you for the introduced revisions and congratulations on good work.